

# Evaluating the role of wild songbirds or rodents in spreading avian influenza virus across an agricultural landscape

Derek D. Houston[1,2], Shahan Azeem[3], Coady W. Lundy[1,4], Yuko Sato[5], Baoqing Guo[5], Julie A. Blanchong[1], Phillip C. Gauger[5], David R. Marks[4], Kyoung-Jin Yoon[3,5] and James S. Adelman[1]

[1] Department of Natural Resource Ecology and Management, Iowa State University, Ames, IA, United States of America
[2] Department of Natural and Environmental Sciences, Western State Colorado University, Gunnison, CO, United States of America
[3] Department of Veterinary Microbiology and Preventive Medicine, Iowa State University, Ames, IA, United States of America
[4] Animal and Plant Health Inspection Service, Wildlife Services, United States Department of Agriculture, Urbandale, IA, United States of America
[5] Department of Veterinary Diagnostic and Production Animal Medicine, Iowa State University, Ames, IA, United States of America

Corresponding authors
Kyoung-Jin Yoon, kyoon@iastate.edu
James S. Adelman, adelmanj@iastate.edu

## ABSTRACT

**Background**. Avian influenza virus (AIV) infections occur naturally in wild bird populations and can cross the wildlife-domestic animal interface, often with devastating impacts on commercial poultry. Migratory waterfowl and shorebirds are natural AIV reservoirs and can carry the virus along migratory pathways, often without exhibiting clinical signs. However, these species rarely inhabit poultry farms, so transmission into domestic birds likely occurs through other means. In many cases, human activities are thought to spread the virus into domestic populations. Consequently, biosecurity measures have been implemented to limit human-facilitated outbreaks. The 2015 avian influenza outbreak in the United States, which occurred among poultry operations with strict biosecurity controls, suggests that alternative routes of virus infiltration may exist, including bridge hosts: wild animals that transfer virus from areas of high waterfowl and shorebird densities.

**Methods**. Here, we examined small, wild birds (songbirds, woodpeckers, etc.) and mammals in Iowa, one of the regions hit hardest by the 2015 avian influenza epizootic, to determine whether these animals carry AIV. To assess whether influenza A virus was present in other species in Iowa during our sampling period, we also present results from surveillance of waterfowl by the Iowa Department of Natural Resources and Unites Stated Department of Agriculture.

**Results**. Capturing animals at wetlands and near poultry facilities, we swabbed 449 individuals, internally and externally, for the presence of influenza A virus and no samples tested positive by qPCR. Similarly, serology from 402 animals showed no antibodies against influenza A. Although several species were captured at both wetland and poultry sites, the overall community structure of wild species differed significantly between these types of sites. In contrast, 83 out of 527 sampled waterfowl tested positive for influenza A via qPCR.

**Discussion**. These results suggest that even though influenza A viruses were present on the Iowa landscape at the time of our sampling, small, wild birds and rodents were unlikely to be frequent bridge hosts.

# INTRODUCTION

Avian influenza (AI) is caused by Type A influenza viruses that exist naturally in wild bird populations and can cross the wildlife-domestic animal interface, sometimes leading to widespread epizootics in domestic poultry (*Alexander, 2007*). Such events can prove extremely costly to the commercial poultry industry and enhance the potential for zoonotic spillover into humans (*Rushton et al., 2005*; *Koopmans et al., 2004*). Clinical manifestations of avian influenza virus (AIV) infection can vary and the viruses are classified as highly or low-pathogenic strains (HPAIV and LPAIV, respectively) based on virulence in poultry, with H5 and H7 subtypes being the most common HPAIVs (*Scholtissek et al., 1978*; *Röhm et al., 1996*; *Alexander, 2000*; *Gamblin et al., 2004*; *Thomas & Noppenberger, 2007*; *Das et al., 2010*; *Wille et al., 2013*; *Latorre-Margalef et al., 2014*; *Lu, Lycett & Brown, 2014*). In the spring of 2015, a HPAIV strain of H5N2 subtype caused the most detrimental and costly outbreak in the United States (*Ip et al., 2015*; *United States Department of Agriculture (USDA APHIS), 2015b*; *Arruda et al., 2016*). This epizootic event had a devastating impact on the regional commercial poultry industry, particularly in Iowa where over 30 million chickens were destroyed with an estimated economic impact of at least $1.2 billion (*Decision Innovation Solutions, 2015*; *Greene, 2015*; *United States Department of Agriculture (USDA APHIS), 2015a*). In some cases, initial introduction of AIV from wild bird populations into domestic flocks has been attributed to migratory waterfowl, but in others it has been introduced via human activities or other unknown factors (*Thomas & Noppenberger, 2007*). Given the destructive impacts of HPAI outbreaks it is important to better understand modes of AIV transmission.

Migratory waterfowl and shorebirds are natural reservoirs for AIV (*Webster et al., 1992*; *Olsen et al., 2006*; *Munster et al., 2007*; *Latorre-Margalef et al., 2009*; *Wilcox et al., 2011*; *Huang et al., 2013*; *Jenelle et al., 2016*; but see also *Caron, Cappelle & Gaidet, 2017*). These birds often exhibit few clinical signs of infection, but they can carry and shed the virus along migratory pathways (*Olsen et al., 2006*; *Kilpatrick et al., 2006*; *Lebarbenchon et al., 2012*; *Pepin et al., 2014*; *Fries et al., 2015*; *Bengtsson et al., 2016*), and thus are generally considered to be vital reservoirs for AIV (*Reed et al., 2003*). AIV can infect other species, including terrestrial bird populations (i.e., songbirds, woodpeckers, etc.) and domestic poultry (*Caron, Cappelle & Gaidet, 2017*; *Kilpatrick et al., 2006*; *Sonnberg, Webby & Webster, 2013*). Such infections typically do not result in severe disease outbreaks, but HPAI outbreaks can emerge in domestic flocks if LPAIV strains mutate into HPAIV strains, if multiple LPAIV strains reassort and become HPAIV strains (*Scholtissek et al., 1978*; *Gamblin et al., 2004*; *Thomas & Noppenberger, 2007*; *Wille et al., 2013*; *Lindsay et al., 2013*), or if domestic

poultry are infected with HPAIV from elsewhere (*Capua & Marangon, 2000*). It remains unclear how AIV is transmitted into domestic bird populations, especially considering that most poultry farms in developed nations now enforce strict biosecurity protocols to prevent outbreaks facilitated by human activities (although in practice, compliance may be inconsistent), waterfowl rarely inhabit commercial poultry farms in areas where some outbreaks have occurred, and disease outbreaks spread regionally among domestic populations even after migratory bird movements have ended (*Thomas & Noppenberger, 2007*). Further complicating the issue of AI outbreaks, prior studies have shown that annual prevalence of AIV can be cyclical in wild waterfowl, suggesting that re-emergence of the disease is a threat even after isolated outbreaks have subsided (*Hinshaw et al., 1985*; *Curran et al., 2015*).

Given the severity of the 2015 AIV outbreak, along with Iowa's close proximity to multiple migratory pathways (Iowa is administratively classified under the Central flyway, but birds from the Mississippi and Atlantic flyways pass through some parts of the state), and intense egg production in the state, this region is at high risk for AIV and is thus an important area in which to study potential means of AIV transmission. With uncertainty about the mechanisms of AIV transfer into domestic poultry and the high possibility of AIV re-emergence, we sought to examine alternate conduits of AIV transmission from wildlife reservoirs into domestic poultry farms. Specifically, we performed surveillance on small, wild birds (i.e., non-waterfowl) and mammals as potential bridge hosts for AIV transfer from wetlands to commercial poultry operations (*Kou et al., 2005*; *Nemeth et al., 2010*; *Shriner et al., 2012*; *Wanaratana et al., 2013*; *Zhao et al., 2014*; *Caron et al., 2014*; *Caron et al., 2015*).

We focused on wild bird (non-waterfowl) and small mammal species for the following reasons. First, modern poultry production often occurs in confinements without large openings, effectively barring waterfowl from entering. Second, as part of the biosecurity measures concerning introduction of AIV from waterfowl, many of these facilities are located away from large water bodies such as lakes or ponds commonly used by migratory waterfowl. Third, the most abundant wildlife residing in poultry farms and feed mills are small songbirds and rodents (*Brown et al., 2014*; *Leon et al., 2013*), some of which have been shown capable of carrying AIV experimentally or through surveillance programs (*Jenelle et al., 2016*; *Caron, Cappelle & Gaidet, 2017*; *Kou et al., 2005*; *Nemeth et al., 2010*; *Shriner et al., 2012*; *Zhao et al., 2014*). Fourth, these small wild birds and mammals are capable of travel between poultry barns and wetlands where waterfowl stop during migration, serving as potential bridge hosts that may augment the risk of poultry epizootics (*Caron et al., 2014*; *Caron et al., 2015*; *Caron et al., 2009*; *Caron et al., 2010*). Given the severity of the 2015 AI epizootic in North America and the potential for these small animals to act as conduits between AIV-contaminated wetlands and commercial poultry facilities (*Caron et al., 2014*; *Caron et al., 2015*), it is important to understand the roles, if any, that non-waterfowl wild birds and mammals play in spreading the virus.

Although some prior studies have reported AIV in atypical reservoir or vector species, such as songbirds and small mammals (*Slusher, 2013*; *Reperant, Rimmelzwaan & Kuiken, 2009*), such work often omits critical considerations about AIV biology and salient sampling

locations (*Hoye et al., 2010*). First, prior studies of AIV in small terrestrial birds have focused almost exclusively on successful infection of these species (i.e., detecting AIV *inside* an animal) (*Slusher, 2013*). This type of study, while informative, neglects an important aspect of AIV biology: these viruses can persist outside of the body (*Beigel et al., 2005*) and could be transmitted mechanically (i.e., on the *outside* of an animal) (*Brown et al., 2014*; *Kaleta & Honicke, 2004*). As such, the ability of small, wild birds to transfer AIV from conventional wildlife reservoirs (e.g., waterfowl) into commercial poultry facilities may be underestimated. Second, persistence of AIV outside of an avian host leaves open the possibility that other animals, such as rodents, could also transport AIV, either internally or externally (*Shriner et al., 2012*; *Wanaratana et al., 2013*). Third, prior studies of AIV in songbirds or mammals have often included habitat types with little or no potential for interaction among species of concern (i.e., waterfowl), thus missing or diluting the most important sampling locations (*Peterson et al., 2008*; *Siengsanan et al., 2009*; *Fuller et al., 2010*; *Thinh et al., 2012*; but see also *Zhao et al., 2014*; *Caron et al., 2014*; *Leon et al., 2013*; *Peterson et al., 2008*; *Siengsanan et al., 2009*; *Fuller et al., 2010*). In contrast, ideal sampling should focus on habitats where potential bridge hosts, including small mammals and birds, are most likely to interact with known AIV reservoirs like migratory waterfowl and shorebirds (e.g., wetlands and marshes) and to interact with poultry or their feed (e.g., commercial poultry operations, or feed-mills that serve those operations) (*Caron et al., 2014*; *Caron et al., 2015*; *Gronesova et al., 2008*; *Borovská et al., 2011*; *Cumming et al., 2011*). Hence, the actual role of small birds and mammals in spreading AIV has not been definitively evaluated, particularly in the United States, even though these species have the potential to carry AIV biologically and mechanically.

While surveillance among these types of species will help determine their potential to carry AIV, successful bridge species must also have the potential to visit both wetland sites and poultry facilities (*Caron et al., 2014*; *Caron et al., 2015*). As such, assessing the risk of small birds and mammals as potential bridge species requires some consideration of community structure (i.e., the types and abundances of species present) at different types of sites (*Caron et al., 2014*; *Caron et al., 2015*). For instance, if a given species is found to carry AIV, but inhabits wetlands exclusively and never visits poultry facilities, that species is unlikely to successfully facilitate AIV transmission from wild to domestic animals.

Our objective in this study, conducted in the wake of the 2015 AIV outbreak in the United States, was to evaluate the potential of small wild birds and rodents to serve as bridge species for AIV transmission. To do so, we assessed the prevalence of AIV in a variety of wild birds and rodents (internally or externally) using qPCR and serology, and compared species communities captured at sites near wetlands vs. commercial poultry facilities across Iowa, USA. To determine whether influenza A viruses were present on the Iowa landscape more generally during our sampling, we compared our results with those from sampling efforts in waterfowl performed under a separate effort by the Iowa Department of Natural Resources and US Department of Agriculture Wildlife Services.

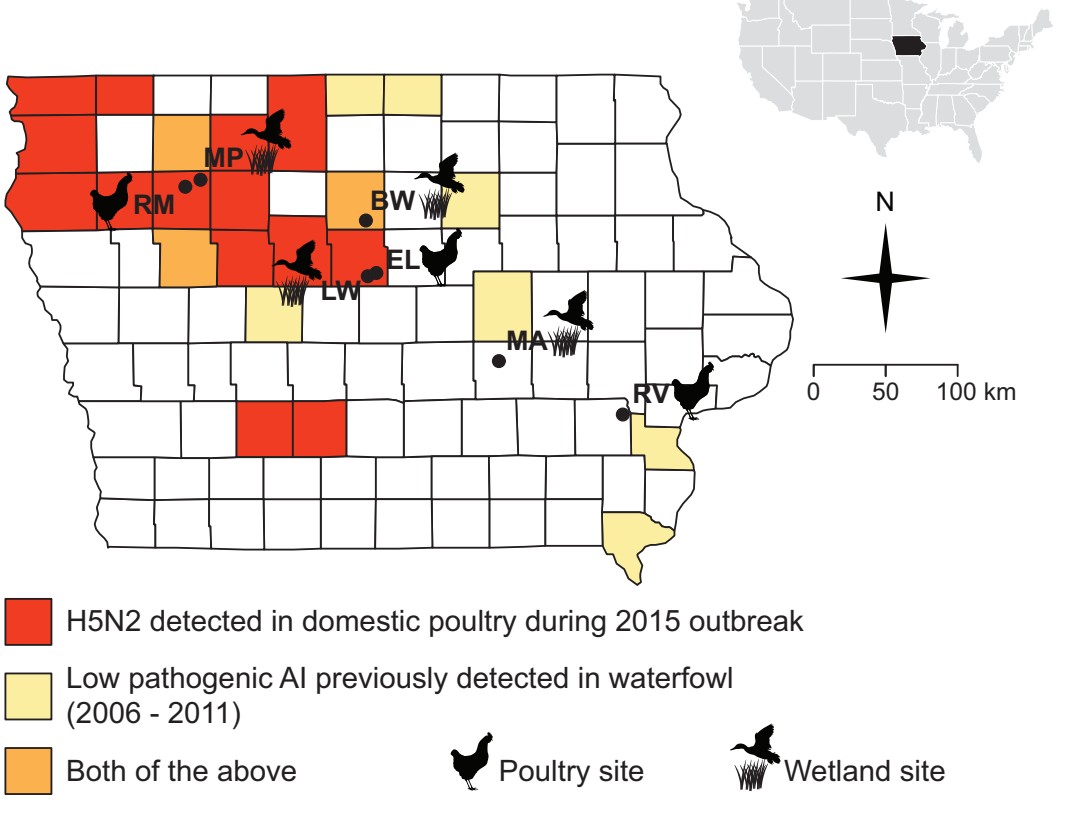

**Figure 1** Sampling sites were chosen in counties where the 2015 H5N2 outbreak occurred (red, orange) or along a diagonal band where prior surveillance had found AIV in waterfowl (yellow, orange). Sampling localities visited for this study (during Fall 2015 and Spring 2016) are marked with black circles and names are abbreviated as follows: Big Wall Lake (BW), Ellsworth (EL), Little Wall Lake (LW), Malcom (MA), Marathon Poland Park (MP), Rembrandt (RM), Riverside (RV).

## MATERIALS AND METHODS

### Field sampling, small birds and mammals

The Iowa State University Animal Care and Use Committee approved all procedures for the handling of specimens and samples (Protocol 9-15-8094-W). Field collections and captures were approved under the following state and federal permits: Iowa Department of Natural Resources Scientific Collecting Permit (SC1133 to JSA), United States Geological Survey Bird Banding Lab Master Banding Permit (23952 to JSA). We obtained samples from wild birds and small mammals at seven sites distributed across Iowa, USA (Fig. 1; Table 1). Samples were collected during and after the fall migration of 2015 (October 23–December 14), and during spring migration of 2016 (March 22–May 10), when the movements of waterfowl would be most likely to bring AIV to the region and when ambient temperatures are low enough to allow persistence of AIV in the environment (*Handel et al., 2014*; *Dalziel et al., 2016*). Sampling sites (Fig. 1) were chosen based on their proximity to areas that experienced HPAI outbreaks in 2015 or where prior monitoring (2006–2011) (*Bevins et al., 2014*) detected AIV in waterfowl (J Baroch, USDA-APHIS-WS, pers. comm. to KJY,

**Table 1  List of sampling localities during fall 2015 and spring 2016.** Numbers of animals sampled from each locality are listed for each season. The first number represents the sample size included in qPCR analysis; the number in parentheses represents the sample size included for serology. Discrepancies reflect individuals from which blood samples were not taken due to escape or insufficient blood draw. Recaptured animals were immediately released and not sampled a second time.

| | Fall 2015 | | Spring 2016 | | |
| Sampling locality | Birds | Mammals | Birds | Mammals | Total |
|---|---|---|---|---|---|
| Big Wall Lake[w] | 23 (21) | 25 (15) | 25 (24) | 10 (9) | 83 (69) |
| Ellsworth[P] | 27 (25) | 22 (19) | 29 (29) | 10 (10) | 88 (83) |
| Little Wall Lake[w] | 19 (18) | 33 (18) | 21 (20) | 1 (0) | 74 (56) |
| Malcom[w] | 18 (17) | 21 (21) | 27 (26) | 11 (7) | 77 (71) |
| Marathon Poland Park[w] | 14 (14) | 16 (16) | 42 (42) | 2 (2) | 74 (74) |
| Rembrandt[P] | 0 (0) | 0 (0) | 21 (18) | 0 (0) | 21 (18) |
| Riverside[P] | 0 (0) | 0 (0) | 31 (30) | 1 (1) | 32 (31) |
| Total | 101 (95) | 117 (89) | 196 (189) | 35 (29) | 449 (402) |

**Notes.**
[P] Denotes a domestic poultry farm.
[w] Denotes a wetland site.

2014). Four of the sites were wetlands (Big Wall Lake, Little Wall Lake, Malcom, Marathon Poland Park), three of which are located in counties where HPAI outbreaks occurred in 2015, and the remaining three sites were commercial properties (Ellsworth, Rembrandt, Riverside), two of which were in counties that experienced HPAI occurrence during 2015. Precise choices of sampling sites were constrained by cooperation with local landowners and poultry producers concerned with the additional surveillance our sampling would represent on their property. Waterfowl were observed at all wetland sites during the fall and spring sampling periods (by DDH and CL). The time spent sampling each site averaged 4.9 days in the fall and 3.1 days in the spring, with time spent sampling per day ranging from 8 to 12 h.

For capture, we targeted small bird and mammal species that spend significant time on the ground, where they were most likely to interact with waterfowl or wading birds. Small wild birds were captured using mist nets deployed near bird feeders that had been placed at sites between two and five days prior to sampling. At wetland sites, nets were placed within 100 m of water; at poultry farms, nets were placed as close to buildings as cooperators would permit, with a range of 10–200 m. In addition, some samples of invasive avian species (house sparrows (*Passer domesticus*) and European starlings (*Sturnus vulgaris*), $N = 9$) were obtained by lethal collection via air rifle. Small mammals were trapped using folding, metal live-traps (H.B. Sherman Traps, Inc., Product #: LFA, Tallahassee, FL, USA). We deployed 200 traps at each site. Mammal traps were placed on the ground within 100 m of water at wetland sites and between 10–200 m of buildings at poultry facilities. Traps were baited with peanut butter, which was wrapped in waxed paper and frozen prior to deployment. Traps were placed at dusk. On nights with projected overnight temperatures below 40 °F, traps were lined with cotton balls for any trapped animals to use for insulation.

At both wetland and commercial poultry sites, we began netting birds and checking mammal traps between 6–8 am and continued through the daylight hours, weather

permitting. Nets and traps were closed if rain or snow became heavy, but left open in light drizzle or flurries. In such cases, nets were checked and any captured animals were removed every 5–10 min.

Three samples were obtained from each animal captured. First, individuals were swabbed externally (e.g., feet, feathers/fur) with single-use, sterile polyester fiber-tipped synthetic swabs, which were placed into individually labeled tubes containing 2 mL of brain heart infusion (BHI) medium, which has been demonstrated to be effective for AIV recovery (*Spackman et al., 2013*). Next, internal samples were taken with oropharyngeal and cloacal/anal swabs, and the two internal swabs from each individual were pooled into a single labeled tube containing 2 mL of BHI. Both oropharyngeal and cloacal/anal swabs were taken because it has been demonstrated that different species exhibit different levels of AIV in these swabs, suggesting variation among species, or virus strains, in potential transfer (*Costa et al., 2011*). Lastly, blood samples were taken using heparinized microhematocrit tubes following venipuncture of the brachial vein (wing) using 26 or 27 gauge needles for birds, or the saphenous vein (leg) using a 23 gauge needle for mammals (after removing leg hair using electric clippers). Blood samples were immediately transferred to individually labeled 0.7 mL microcentrifuge tubes, free of any additional anticoagulant. All BHI and blood samples were chilled during transport, and were usually back in the laboratory within 8 h of collection. All individuals were released at their point of capture after processing was completed; and birds were banded prior to their release. Recaptured individuals identified by bands (birds) or by the presence of a shaved patch of fur (mammals), were immediately released without resampling. Upon arrival at the laboratory, all blood samples were immediately spun in mini-centrifuges for 10 min to separate red blood cells from plasma, then plasma was transferred to new individually labeled microcentrifuge tubes using a 100 µL Hamilton syringe. All blood, plasma, and BHI samples were stored at $-20\,^{\circ}\text{C}$ until they were transferred in an Iowa State University vehicle, still frozen, to the Veterinary Diagnostic Laboratory (VDL) at Iowa State University for further processing. Transfer to the VDL occurred one week after the end of each sampling season, each of which spanned eight weeks, so individual samples were frozen for 1–9 weeks before transfer.

### Field sampling, waterfowl

From August 2015 through January 2016, a separate team of researchers from the Iowa Department of Natural Resources (DNR) and the United States Department of Agriculture (USDA) Wildlife Services (WS) collected 527 samples of waterfowl, both from hunter-collected carcasses and by live-trapping. Data presented here are done so in accordance with USDA data transfer policy. Oropharyngeal and cloacal swabs were collected and pooled by individual (making one sample per individual) to assess the presence of virus via qPCR (see below), but blood samples were not drawn to assess serology. Samples were collected from several Iowa counties: Adair, Appanoose, Cerro Gordo, Hancock, Harrison, Jackson, Johnson, Louisa, Lucas, Marshall, Union, Wayne, and Winneshiek (Fig. 2). Swabs from waterfowl were placed into BHI medium and stored on ice until freezing at $-20\,^{\circ}\text{C}$ before lab processing. Samples were collected, processed and shipped following the protocols previously described (*DeLiberto et al., 2009*; *Pedersen, Swafford & DeLiberto, 2010*).

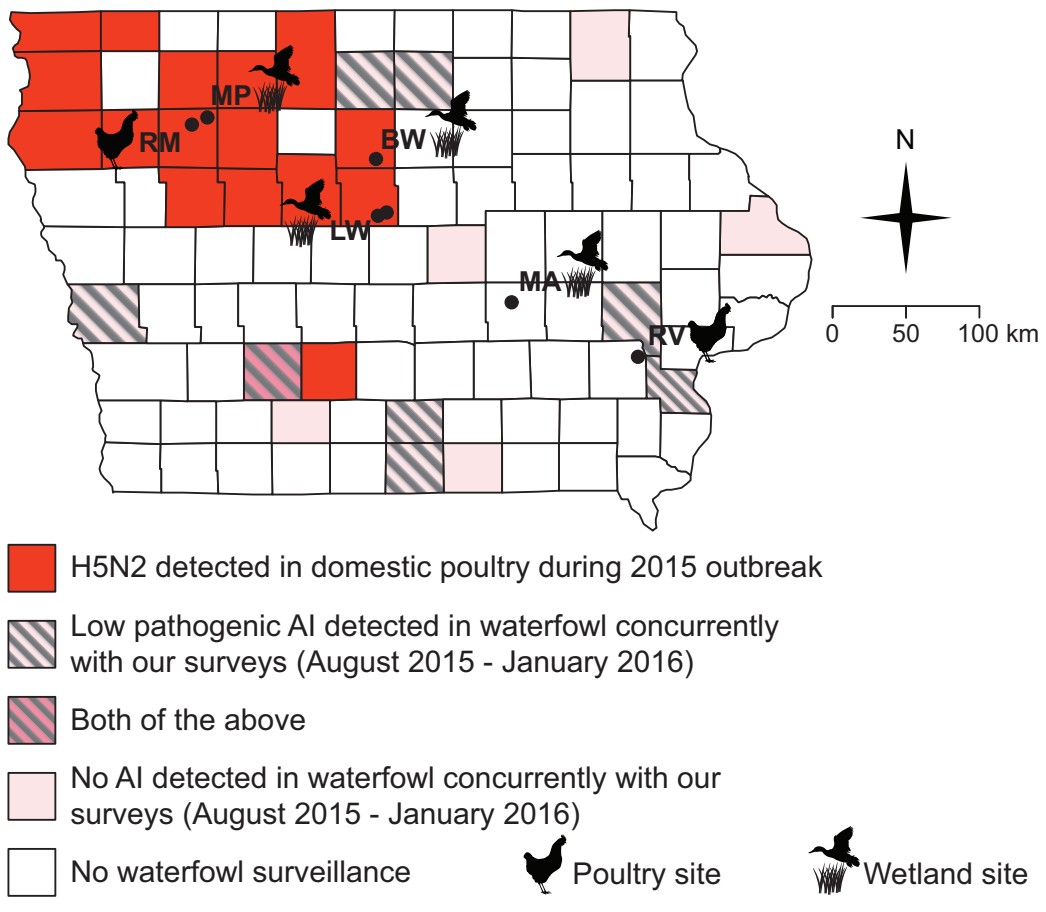

**Figure 2** **All but one of our sampling sites fell within counties impacted by the 2015 H5N2 outbreak or adjacent to counties where low pathogenic AIV was detected in waterfowl between August 2015 and January 2016.** Black circles and abbreviated site names (see Fig. 1) indicate locations where we sampled small birds and mammals during this study (October–December, 2015; March–May 2016).

## Laboratory testing, small birds and mammals
### Sample processing

All samples were processed as soon as possible upon arrival to the veterinary diagnostic laboratory, or otherwise stored at −80 °C. All laboratory work was performed at Biosafety Level 2 (BSL2) compliance and any work involving virus isolates occurred in a laminar flow biosafety cabinet. All swab tubes containing BHI medium were vortexed for 60 s and an aliquot of BHI medium from each tube was dispensed into 96-well plates for RNA extraction. No additional processing other than aliquoting plasma samples was required prior to testing for the virus or anti-AIV antibodies.

### PCR assay for influenza A virus

We extracted viral RNA from each swab sample via magnetic bead based separation technology using an Ambion® MagMAX™-96 Viral RNA Isolation Kit (Life Technologies, Carlsbad, CA, USA) following the protocol provided by the manufacturer. The procedure was performed in a KingFisher® 96 automated magnetic particle processer (ThermoFisher

Scientific, Prussia, PA, USA) as per manufacturer's instructions. Extracted viral RNA was eluted in 50-$\mu$L elution buffer.

We used a commercially available one-step real-time multiplex RT-PCR kit (VetMAX$^{TM}$ Gold AIV Detection Kit; Life Technologies, Austin, TX, USA), designed to target viral matrix and nucleoprotein genes, to amplify influenza viral RNA. The USDA approves this kit for AIV surveillance testing. The PCR reaction was set up in a 25 $\mu$L volume containing 12.5 $\mu$L of 2X multiplex RT-PCR buffer, 1.0 $\mu$L nuclease-free water, 1.0 $\mu$L of influenza virus primer probe mix, 2.5 $\mu$L of multiplex RT-PCR enzyme mix and 8.0 $\mu$L of RNA template (i.e., extract) or controls. Xeno$^{TM}$ RNA Control supplied with the kit was included as an internal control for RNA purity to assess possible PCR inhibition from samples. Influenza Virus-Xeno$^{TM}$ RNA Control (1,000 copies/$\mu$L) included in the kit was used as a positive amplification control (PAC). A low-pathogenic AIV isolate (A/Turkey/WI/68 (H5N9)), which was obtained from the USDA, was used as an AIV Matrix PCR extraction control. Nuclease-free water was used as a no amplification control. Thermocycling was performed in a 7,500 Fast PCR System (Applied Biosystems, Foster City, CA, USA) under the following conditions: reverse transcription at 48 °C for 10 min, reverse transcriptase inactivation/initial denaturation at 95 °C for 10 min, and 40 cycles of amplification and extension (95 °C for 15 s and 60 °C for 45 s) (*Zhang & Harmon, 2014*).

We analyzed the PCR data using "Manual Cycle Threshold ($C_T$)" and default baseline cycle 3–15. The AIV master detector threshold was determined by multiplying the delta Rn of PAC at cycle 40 by 0.05. Amplification plots were viewed to ensure that positive controls crossed the threshold and that negative controls did not. AIV RNA and Xeno$^{TM}$ RNA control were detected by using FAM$^{TM}$ and VIC$^{TM}$ dyes, respectively. Samples with $C_T$ values $\leq$40 were recorded as positive for influenza A viral RNA, whereas samples with $C_T$ values >40 were recorded as negative as per manufacturer's instructions.

### Serology

Plasma samples were tested on a USDA-approved IDEXX AI MultiS-Screen Ab Test kit (IDEXX Laboratories, Inc. Westbrook, ME, USA) for influenza A virus antibodies. In brief, samples were vortexed before transferring them to test plates, 100 $\mu$L of undiluted negative and positive kit controls each, as well as 100 $\mu$L of 1:10 diluted house controls or plasma samples were dispensed into appropriate wells of the plate, and plates were incubated for 60 min. Each plate was washed 3–5 times with the wash solution that was supplied with the kit, after which 100 $\mu$L of anti-AI horseradish-peroxidase conjugate was dispensed into each well and the plate was incubated for another 30 min. After incubation, the plates were washed again, then 100 $\mu$L of substrate solution was dispensed into each well and the plates were incubated for another 15 min. The reaction was stopped by addition of 100 $\mu$L of stop solution and absorbance (i.e., optical density) was read at a 650 nm wavelength. All incubation steps were conducted at ambient temperature.

Each plate was validated by noting the absorbance of the negative and positive control means, and, if validated, the Sample/Negative (S/N) ratio was calculated for each sample. In accordance with manufacturer recommendations, if the S/N ratio was <0.5, the sample was classified as positive for antibodies to influenza A virus. If the S/N ratio was $\geq$ 0.5, the

sample was classified as negative for antibodies to influenza A virus. We note that if a more conservative cutoff value of 0.6 is applied, our results do not change.

## Laboratory testing, waterfowl

Waterfowl samples were tested at the United States Geological Survey National Wildlife Health Center (NWHC) in Madison, Wisconsin. The NWHC is one of the National Animal Health Laboratory Network facilities and is certified by the USDA National Veterinary Services Laboratories. Samples were tested following the national wild bird surveillance program protocols, including qPCR matrix tests and additional qPCR to probe for H5 and H7 if matrix tests proved positive. Detailed descriptions of the national wild bird surveillance diagnostic testing protocols have been described previously (*DeLiberto et al., 2009*; *DeLiberto, et al., 2011*).

### *Analyses*

When estimating disease prevalence, frequentist statistical approaches assume perfect detection in the laboratory assays. Because this is unlikely, we estimated the prevalence of AIV infection and exposure in potential bridge species using a Bayesian approach that incorporates estimates of assay sensitivity and specificity (*Williams & Moffitt, 2010*) in R v.3.1.3 (*R Development Core Team, 2015*). We used the following values for assay sensitivity and specificity, obtained from the manufacturers and based upon samples from waterfowl, with a possible range from 0–1: ELISA sensitivity = 0.820, ELISA specificity = 1.00; qPCR sensitivity = 0.984, qPCR specificity = 0.991. It should be noted, that although these tests have been utilized previously in myriad species, they were originally developed for use in waterfowl and domestic poultry, so manufacturer's sensitivities and specificities may not reflect performance in passerines or mammals. In addition, because our analysis method incorporates imperfect detection, there is always some chance, however small, of false negatives; our confidence intervals therefore do not include absolute 0% prevalence. We report only raw results for waterfowl sampling because these data are included for the sole purpose of illustrating that influenza A viruses were present in Iowa around the time of our sampling. As such, our goal was not to provide any estimate of prevalence in these species.

Communities of small birds and mammals captured (number and abundances of different species) were compared between wetland and poultry sites using the vegan package v.2.3-5 in R (*Oksanen et al., 2016*). Briefly, community similarity was calculated between each site using Bray-Curtis distance estimation and analyzed using permutation-based ANOVA (PERMANOVA). This analysis was conducted including all species and for birds only. Analyses were not performed on mammals alone, as so few species were captured. In these analyses, we assume, as have several prior studies, that the animals captured reflect an accurate sample of the community present. We note, however, some species present were likely not captured, particularly high-flying avian species (*Bonter, Brooks & Donovan, 2008*).

## RESULTS

Samples from a total of 449 wild birds and small mammals were obtained from four wetland sites and three domestic poultry farms distributed across Iowa (Figs. 1 and 2, Tables 1 and 2. None of these 449 individuals tested positive for influenza A virus by qPCR from external

**Table 2   Potential AIV bridge species sampled by species and site type (poultry-adjacent or wetland-adjacent).** Individuals were sampled at four wetland sites and three domestic poultry farms in Iowa during the Fall 2015 and Spring 2016 sampling seasons. Numbers in parentheses indicate the percentages of all individuals captured at a given site type that belong to a given species. All individuals were swabbed externally (feet, feathers/fur) and internally (oropharyngeal, anal/cloacal) to test for the presence of AIV. We were able to collect blood samples from the majority of individuals (402/449) to test for presence of anti-AIV antibodies.

| Species | | Num. individuals, poultry sites (%) | Num. individuals, wetland sites (%) | Num. individuals, total (%) |
|---|---|---|---|---|
| **Birds** | | | | |
| Dark-eyed junco[a] | *Junco hyemalis* | 22 (15.6%) | 75 (24.4%) | 97 (21.6%) |
| House sparrow | *Passer domesticus* | 44 (31.2%) | 0 (0%) | 44 (9.8%) |
| Song sparrow | *Melospiza melodia* | 0 (0%) | 22 (7.1%) | 22 (4.9%) |
| American tree sparrow[a] | *Spizelloides arborea* | 1 (0.7%) | 20 (6.5%) | 21 (4.7%) |
| American robin[a] | *Turdus migratorius* | 13 (9.2%) | 7 (2.3%) | 20 (4.5%) |
| Red-winged blackbird | *Agelaius phoeniceus* | 0 (0%) | 13 (4.2%) | 13 (2.9%) |
| Northern cardinal | *Cardinalis cardinalis* | 0 (0%) | 11 (3.6%) | 11 (2.4%) |
| Common grackle[a] | *Quiscalus quiscula* | 1 (0.7%) | 9 (2.9%) | 10 (2.2%) |
| Black-capped chickadee | *Poecile atricapillus* | 0 (0%) | 6 (1.9%) | 6 (1.3%) |
| European starling | *Sturnus vulgaris* | 5 (3.5%) | 0 (0%) | 5 (1.1%) |
| Fox sparrow | *Passerella iliaca* | 0 (0%) | 5 (1.6%) | 5 (1.1%) |
| Blue jay | *Cyanocitta cristata* | 0 (0%) | 4 (1.3%) | 4 (0.9%) |
| Chipping sparrow | *Spizella passerina* | 4 (2.8%) | 0 (0%) | 4 (0.9%) |
| White-throated sparrow | *Zonotrichia albicollis* | 4 (2.8%) | 0 (0%) | 4 (0.9%) |
| Eastern phoebe[a] | *Sayornis phoebe* | 2 (1.4%) | 1 (0.3%) | 3 (0.7%) |
| Rusty blackbird[a] | *Euphagus carolinus* | 1 (0.7%) | 2 (0.6%) | 3 (0.7%) |
| White-crowned sparrow[a] | *Zonotrichia leucophrys* | 2 (1.4%) | 1 (0.3%) | 3 (0.7%) |
| American goldfinch[a] | *Spinus tristis* | 1 (0.7%) | 1 (0.3%) | 2 (0.4%) |
| Brown-headed cowbird[a] | *Molothrus ater* | 1 (0.7%) | 1 (0.3%) | 2 (0.4%) |
| Brown thrasher[a] | *Toxostoma rufum* | 1 (0.7%) | 1 (0.3%) | 2 (0.4%) |
| Rock pigeon | *Columba livia* | 2 (1.4%) | 0 (0%) | 2 (0.4%) |
| Swamp sparrow | *Melospiza georgiana* | 0 (0%) | 2 (0.6%) | 2 (0.4%) |
| Wood thrush | *Hylocichla mustelina* | 2 (1.4%) | 0 (0%) | 2 (0.4%) |
| Baltimore oriole | *Icterus galbula* | 1 (0.7%) | 0 (0%) | 1 (0.2%) |
| Brewer's blackbird | *Euphagus cyanocephalus* | 0 (0%) | 1 (0.3%) | 1 (0.2%) |
| Brown creeper | *Certhia americana* | 0 (0%) | 1 (0.3%) | 1 (0.2%) |
| Downy woodpecker | *Picoides pubescens* | 0 (0%) | 1 (0.3%) | 1 (0.2%) |
| Eastern bluebird | *Sialia sialis* | 0 (0%) | 1 (0.3%) | 1 (0.2%) |
| Golden-crowned kinglet | *Regulus satrapa* | 0 (0%) | 1 (0.3%) | 1 (0.2%) |
| Harris's sparrow | *Zonotrichia querula* | 0 (0%) | 1 (0.3%) | 1 (0.2%) |
| Ring-necked pheasant | *Phasianus colchicus* | 1 (0.7%) | 0 (0%) | 1 (0.2%) |
| White-breasted nuthatch | *Sitta carolinensis* | 0 (0%) | 1 (0.3%) | 1 (0.2%) |
| **Mammals** | | | | |
| Deer mouse[a] | *Peromyscus sp.* | 3 (2.1%) | 109 (35.4%) | 112 (24.9%) |
| House mouse[a] | *Mus musculus* | 19 (13.5%) | 1 (0.3%) | 20 (4.5%) |
| Northern short-tailed shrew[a] | *Blarina brevicauda* | 5 (3.5%) | 6 (1.9%) | 11 (2.4%) |

**Table 2** (*continued*)

| Species | | Num. individuals, poultry sites (%) | Num. individuals, wetland sites (%) | Num. individuals, total (%) |
|---|---|---|---|---|
| Meadow vole[a] | *Microtus pennsylvanicus* | 2 (1.4%) | 2 (0.6%) | 4 (0.9%) |
| Norway rat | *Rattus norvegicus* | 4 (2.8%) | 0 (0%) | 4 (0.9%) |
| Long-tailed weasel | *Mustela frenata* | 0 (0%) | 1 (0.3%) | 1 (0.2%) |
| **Totals** | | 141 | 308 | 449 |

**Notes.**
[a]Species occurs at both poultry and wetland sites.

or internal swabs (95% CI of prevalence [0.005%–0.83%]; note that because our analyses account for less than 100% sensitivity, confidence intervals do not include 0). Serology was possible on blood samples from 402 animals, none of which showed antibodies against influenza A virus (95% CI of prevalence [0.01%–1.21%]). Our sample sizes were more modest for any one species and as such, estimates of prevalence for specific species are considerably less precise. For qPCR estimates, these would range from 0.03%–3.75% for the most abundant species (dark-eyed juncos–note "*Peromyscus sp.*" included more individuals, but of two difficult-to-distinguish species lumped together, *P. maniculatus* and *P. leucopus*) to 1.39%–85.2% for the least abundant species (any of those with only 1 individual). Similarly, serology-based estimates would range from 0.04%–4.61% for dark-eyed juncos to 1.37%–91.6% for those species with only one individual captured.

Overall species community composition of species captured showed differences between wetland-adjacent and poultry-adjacent sites (PERMANOVA, all species: $F_{1,5} = 5.36$, $p = 0.026$; bird species only: $F_{1,5} = 3.94$, $p = 0.025$). However, there was overlap in community composition, with 14 out of 39 species captured at both types of sites (Table 2), including six of the 10 most commonly captured species (three bird and three mammal species).

Surveillance of waterfowl conducted by the USDA WS and Iowa DNR from August of 2015 through January of 2016 show that avian influenza was present on the Iowa landscape during our sampling (Fig. 2, Table 3). Of 527 samples collected from waterfowl, 83 tested positive for avian influenza A virus by matrix qPCR, with 20 testing positive for H5 subtypes and none testing positive for H7 subtypes by additional, specific qPCRs (Table 3). Virus was not isolated from any of these samples. Positives were spread among eight out of 13 counties sampled across the state (Fig. 2).

## DISCUSSION

We found no evidence for low- or high-pathogenic AIV in small wild birds or mammals across a predominantly agricultural landscape in two migratory seasons following an AIV epizootic. None of the 449 individuals we sampled carried AIV internally or externally based on our qPCR results. Moreover, no influenza A-specific antibodies were detected in the 402 serological samples, suggesting these animals had not been recently exposed. It remains unclear how long anti-AIV antibodies persist in small mammals and birds. However, surveillance in wild geese suggest that while antibody levels can wane across seasons, they remain detectable for at least 3–6 months (*Hoye et al., 2011*; *Samuel et al.,*

**Table 3 Waterfowl sampled by USDA-WS/IADNR indicate that influenza A was present on the Iowa landscape during 2015–2016.** Samples reflect pooled cloacal and oropharyngeal swabs collected from both live-captured and hunter-harvested individuals from 13 Iowa counties between August 2015 to January 2016.

| Species | | Total num. sampled | Num. positive for influenza A (matrix qPCR) | Num. positive for H5 qPCR | Num. positive for H7 qPCR |
|---|---|---|---|---|---|
| American green-winged teal | *Anas carolinensis* | 17 | 1 | 0 | 0 |
| American wigeon | *Anas americana* | 3 | 0 | 0 | 0 |
| Blue-winged teal | *Anas discors* | 63 | 9 | 1 | 0 |
| Mallard | *Anas platyrhynchos* | 206 | 70 | 17 | 0 |
| Northern pintail | *Anas acuta* | 7 | 0 | 0 | 0 |
| Northern shoveler | *Anas clypeata* | 11 | 3 | 2 | 0 |
| Redhead | *Aythya americana* | 1 | 0 | 0 | 0 |
| Wood duck | *Aix sponsa* | 219 | 0 | 0 | 0 |
| **Totals** | | 527 | 83 | 20 | 0 |

*2015*), although it is possible such levels result from repeated exposure. Given that the HPAI outbreaks occurred in summer 2015, we expected to detect antibodies in wild birds or mammals at least during our fall 2015 sampling if these animals had been exposed. That we did not detect AIV in any of our samples via qPCR or anti-AIV antibodies in serology suggests that infection was highly unlikely in small birds and mammals in Iowa at the time of sampling, consistent with most prior surveillance in these types of animals (*Jenelle et al., 2016*; *Kou et al., 2005*; *Shriner et al., 2012*; *Zhao et al., 2014*; *Grear et al., 2017*).

The lack of AIV positive samples among small birds and rodents cannot be explained by a complete absence of AIV in Iowa during our sampling (Fig. 2). Because surveillance by state and federal agencies detected AIV in waterfowl during the fall of 2015, we can be confident that some amount of virus was present in the state. However, because this surveillance was not conducted in a randomized design, estimates of overall prevalence for AIV in the state's waterfowl would not be robust. Moreover, we must note that surveillance for AIV in waterfowl did not completely overlap (spatially or temporally) with our sampling of small birds and rodents (Fig. 2). As such, it is possible that AIV was only present briefly in a select few locations in Iowa, which contributed to our lack of positive samples in small birds and rodents. However, the fact that 8/13 sampled counties showed positive samples from waterfowl suggests that this is unlikely. Additionally, because several waterfowl positives were located in counties adjacent to our small bird and rodent sampling (Fig. 2), it is unlikely that a complete absence of the virus on the Iowa landscape drove the patterns presented here.

Although some species of small birds and mammals are found at both wetland sites and poultry facilities, the overall community structure of these small birds and mammals differs between these types of sites. Taken together with our disease surveillance results, this suggests that on the whole, small, wild birds and mammals are unlikely to play major, ongoing roles in transporting AIV from waterfowl to domestic poultry. However, these community data provide an important set of potential species for further surveillance in the future. Specifically, species that were found at both wetland and poultry sites (Table 2)

are those with the most potential to act as bridge species (*Caron et al., 2014*; *Caron et al., 2015*). Future sampling, with capture techniques targeted toward these species, could improve estimation of AIV prevalence in these animals and our understanding of their role as bridge species. We also note that because we have analyzed communities from samples across the state, it is possible that our estimates of co-occurrence at poultry and wetland sites would vary within specific sub-regions. Moreover, sampling techniques like bird point counts could provide additional information about broader avian communities and may reveal new potential bridge species (*Caron et al., 2014*; *Caron et al., 2015*). We suggest that future surveillance efforts in the Midwest US take such community ecology into account if conducting surveillance in small birds and rodents.

If wild songbirds and small mammals were major sources of AIV transmission to domestic flocks from wetland sites, we would expect these animals to be exposed regardless of whether or not outbreaks were ongoing in commercial operations. However, our data do not suggest that this is the case: we found no evidence of viral RNA or antibodies despite a ∼16% LPAIV infection rate of waterfowl in the region at the time we collected our samples. Indeed, when researchers examine wild populations of small birds and mammals, they typically have only detected AIV in a small proportion of individuals sampled, and detection was more likely during active AIV outbreaks (*Jenelle et al., 2016*; *Kou et al., 2005*; *Shriner et al., 2012*; *Zhao et al., 2014*).

The pattern of small birds and mammals exhibiting low levels of AIV during an epizootic holds true in other surveillance efforts performed during the 2015 HPAI outbreak in the US. *Jenelle et al. (2016)* found very low levels of AIV by PCR in non-waterfowl in Minnesota, US, just north of the areas of Iowa hit hardest during the AI outbreak. Specifically, they isolated HPAIV from one Cooper's hawk, that they postulated had likely been infected via a prey item (*Jenelle et al., 2016*). They also sampled three chickadees that exhibited erratic behavior, but the virus was not isolated, despite HPAI viral RNA being detected in one of them (*Jenelle et al., 2016*). Despite these detections, they found no HPAIV in Minnesota waterfowl at the height of the 2015 outbreak in the Midwest (*Jenelle et al., 2016*). Additionally, after the outbreak had subsided, Grear and colleagues (*Grear et al., 2017*) sampled small, wild birds and mammals at three sites in WI during the fall of 2015, finding no animals positive by qPCR and only two deer mice (*Peromyscus sp.*) with antibodies against AIV (both at poultry farms) out of a total of 284 animals sampled. Finally, sampling at Iowa poultry facilities during the 2015 outbreak, *Shriner et al. (2016)* found only one of 648 peridomestic birds and mammals to be qPCR positive for AIV. Combining these results with ours, collected after the HPAI outbreak had subsided, it is plausible that the AIV detections reflect the virus crossing the wildlife-domestic interface into wild birds from infected domestic populations. This is not unprecedented, as AI outbreaks in African and Asian wild birds have been attributed to spread of the disease from infected domestic poultry (see *Dalziel et al., 2016* and references therein).

Even though we found no evidence here that small wild birds and mammals contribute to the spread of AIV, we are unable to conclude that they cannot. First, with truly low prevalence of disease, sample sizes required for precise estimation of prevalence are extremely high. As such, the true risk of infection may be higher than estimated here or in

other recent studies (*Jenelle et al., 2016*; *Grear et al., 2017*; *Shriner et al., 2016*). Moreover, at the level of individual species, our (and others' *Jenelle et al., 2016*; *Grear et al., 2017*; *Shriner et al., 2016*) sample sizes are often quite modest, meaning that confidence of prevalence for each species is low. We note, however, that our total sample sizes of small birds and mammals are comparable to or exceed those of other studies published in the wake of the 2015 HPAI outbreak that arrive at similar conclusions (*Jenelle et al., 2016*; *Grear et al., 2017*; *Shriner et al., 2016*). Second, it is possible that when small birds and mammals become infected, they die quickly, becoming evolutionary dead ends for AIV (*Feare, 2010*; *Kwon et al., 2005*; *Marjuki et al., 2009*; *Marchenko et al., 2012*), and proving difficult to sample. As such, it is possible that any surveillance of these species underestimates true prevalence. Finally, ecological barriers, such as habitat types and distance should help mitigate the spread of AIV by small birds and mammals (*Caron et al., 2014*; *Caron et al., 2015*; *Lam et al., 2012*). However, it is plausible that such species could still help spread the disease to a degree. Importantly, although the virus may occur at extremely low prevalence in non-waterfowl, among highly abundant species like European starlings, low prevalence could translate a significant number of infected individuals. As such, there remains a real risk that initial outbreaks could be triggered by rare infections and propagated by other means, such as farm-to-farm contact. In addition, species likely to be found at both poultry sites (particularly scavengers) could become infected via food items and then transmit AIV back into waterfowl. For instance, some commercial operations dispose of dead birds via on-site outdoor composting (*Greene, 2015*; *United States Department of Agriculture Animal and Plant Health Inspection Service (USDA-APHIS), 2016*). While proper composting protocols state that care should be taken so that no dead birds are exposed, if wild animals access carcasses before the virus is heat-killed, they could potentially become infected prior to interacting with other wild animals on- or off-site. *Jenelle et al. (2016)* reported HPAI in a Cooper's hawk, which although not typically a scavenger, would regularly prey on birds such as rock pigeons and European starlings that commonly frequent poultry farms and could interact with compost bins. We note, however, that this scenario merely outlines one possibility regarding virus spread once an outbreak in domestic flocks has already begun.

## CONCLUSIONS

We estimated very low prevalence of AIV in small, wild bird and mammal populations, supporting the hypothesis that these organisms do not play major roles as bridge species in the transmission of AIV in Iowa. The differences in types and abundances of small wild birds and mammals at wetland vs. poultry facilities further support this notion. Therefore, our results suggest that spread of the virus likely relies on alternative routes of transmission and further research on alternative routes of AIV transmission, including human-mediated transfer (on clothing, equipment, etc.), airborne particulates, and contaminated food/water sources is warranted. However, given the cyclical nature of AIV outbreaks, the large numbers of small birds and mammals on Midwestern landscape in general, and the possibility of disease reemergence, continued surveillance of these species, particularly those species likely to appear at both wetland and commercial poultry operations, may yet improve our understanding of virus ecology.

## ACKNOWLEDGEMENTS

Special thanks goes to the Iowa poultry producers who cooperated on this project. Grace J. Vaziri assisted in the field. Hannah M. Carroll assisted in the field and with creating Fig. 1, and provided comments that helped improve the quality of the manuscript. We are also grateful to Justin Bahl, Alexandre Caron, and one anonymous reviewer for insightful comments that greatly improved this manuscript.

### Funding

Funding for this research was provided by a grant from the Egg Industry Center. Any opinions, findings, and conclusions or recommendations expressed in this material are those of the authors and do not necessarily reflect the views of the Egg Industry Center or Iowa State University. The funders had no role in study design, data collection and analysis, decision to publish, or preparation of the manuscript.

### Grant Disclosures

The following grant information was disclosed by the authors:
Egg Industry Center.
Iowa State University.

### Competing Interests

The authors declare there are no competing interests.

### Author Contributions

- Derek D. Houston conceived and designed the experiments, performed the experiments, wrote the paper, prepared figures and/or tables, reviewed drafts of the paper.
- Shahan Azeem performed the experiments, analyzed the data, wrote the paper, reviewed drafts of the paper.
- Coady W. Lundy performed the experiments, wrote the paper, reviewed drafts of the paper.
- Yuko Sato conceived and designed the experiments, performed the experiments, wrote the paper, reviewed drafts of the paper.
- Baoqing Guo performed the experiments, analyzed the data, reviewed drafts of the paper.
- Julie A. Blanchong and Phillip C. Gauger conceived and designed the experiments, wrote the paper, reviewed drafts of the paper.
- David R. Marks analyzed the data, contributed reagents/materials/analysis tools, wrote the paper, reviewed drafts of the paper.
- Kyoung-Jin Yoon conceived and designed the experiments, analyzed the data, contributed reagents/materials/analysis tools, wrote the paper, reviewed drafts of the paper.
- James S. Adelman conceived and designed the experiments, performed the experiments, analyzed the data, contributed reagents/materials/analysis tools, wrote the paper, prepared figures and/or tables, reviewed drafts of the paper.

## Animal Ethics

The following information was supplied relating to ethical approvals (i.e., approving body and any reference numbers):

The Iowa State University Animal Care and Use Committee approved all procedures for the handling of specimens and samples (Protocol number 9-15-8094-W).

## Field Study Permissions

The following information was supplied relating to field study approvals (i.e., approving body and any reference numbers):

Field collections and captures were approved under the following state and federal permits: Iowa Department of Natural Resources Scientific Collecting Permit (SC1133 to JSA), United States Geological Survey Bird Banding Lab Master Banding Permit (23952 to JSA).

## Data Availability

The raw data has been provided as a Supplemental File.

## Supplemental Information

Supplemental information for this article can be found online at http://dx.doi.org/10.7717/peerj.4060#supplemental-information.

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
