# Peer review of "Evaluating the role of wild songbirds or rodents in spreading avian influenza virus across an agricultural landscape"

_PeerJ, doi:10.7717/peerj.4060_

## Round 0.1 · original submission · Minor Revisions

We invite a revision with strong attention paid to the comments of Reviewers. Specifically,some points in the discussion that should be improved according to the suggestion of Reviewer 3.

Reviewer 1 ·

Basic reporting

The English is clear and professional, the background information and literature are thorough and appropriate, the structure and figures are appropriate. I could not view the raw data file -- it was all symbols for me. I tried on two computers and try to open it as a text file or import it into Excel.

Experimental design

Experimental design: Original primary research is reported, the research question and motivation are clear, the investigation is rigorous and conforms to ethical standards, and the methods are clear and well-described.

Validity of the findings

Validity of the findings: Findings are robust, conclusions are well-supported, and speculation is clearly identified and appropriate.

Additional comments

The manuscript describes a survey of potential bridge hosts (primarily songbirds and small mammals) to investigate the potential role these species might play in transmitting influenza A viruses from reservoir hosts to poultry. The survey is conducted in Iowa at several poultry and wetland sites during both fall and spring after a large poultry outbreak in the state. Samples were tested for influenza A virus and antibodies and no positives were detected.

The manuscript is well-written and very clear throughout. The background information and discussion are very thorough and well documented.

I only have a single comment that I think should be addressed (and even it is relatively minor):
Line 311 - The authors used a cut-off of 0.6 rather than the manufacturer’s recommended cut-off of 0.5. While the higher threshold is appropriate (and supported by Brown et al. 2009, Clin. Vaccine Immunol. and Shriner et al. 2016, J Vir Methods), the authors use the manufacturer’s sensitivity and specificity values in their analysis of a seroprevalence range. Since the manufacturer’s sensitivity and specificity values are based on a cut-off of 0.5, I recommend that the authors should either apply a cut-off of 0.5 to match the manufacturer’s sensitivity and specificity values (this is the easiest solution since it won’t change results) or they should adopt the sensitivity and specificity values that match the 0.6 cut-off (found in the Brown et al. or Shriner et al. papers).

Minor comments
Line 55: The authors might consider replacing “epidemic” with “epizootic” throughout the manuscript.
Line 85: The authors might be a bit more clear about what they mean by “cyclical” in this sentence. In Ref. 35, I believe the cyclical nature of natural infections is associated with underlying immunity to a particular subtype whereas in Ref. 36, I believe the cyclical nature of infection discussed is the seasonal cycle of infection prevalence.
Line 103: I would quibble that this statement may be generally accurate in Iowa, but that natural ponds frequented by waterfowl are not uncommon on or near poultry facilities in other geographic areas (e.g., MN). Also, NPDES ponds attract waterfowl, even in Iowa.
Line 111: Recommend deleting the “of.”
Line 140: Recommend adding the word “a” before the word “given.”
Line 165: The authors might consider changing the “to” to “in.”
Line 181: Use consistent capitalization for bird common names. Consider adding scientific names at first occurrence of species in the text.
Line 216: I would suggest that variation is dependent on both species and viral strain.
Line 221: Did the micro centrifuge tubes contain an anticoagulant (i.e., EDTA or heparin)?
Line 367: The authors might consider noting that none of the APHIS/DNR positives for LPAIV occurred in counties where they sampled bridge hosts. Did any of the 3 APHIS/DNR negative counties overlap the counties where bridge host sampling occurred?
Line 387: Technically the IDEXX assay tests for influenza A specific antibodies and is not specific for avian viruses.
PeerJ does not include funders in the Acknowledgements section.

·

Basic reporting

The paper is clear and the hypothesis is reasonable. I believe that the negative results are important. The background abstract and background for the study are appropriate. However, the material on surveillance in waterfowl appears as an afterthought. The reader is surprised when the section appears in the methods and the results are barely discussed. Either this section should be removed and the context of influenza prevalence in the areas surveyed are established through published literature. Or, properly describe and analyze the results from waterfolw. What subtypes were present? Were any of the samples isolated? Etc.

Experimental design

The design is fine. I think for the results to be truly conclusive longitudinal surveillance is necessary. But this does provide some important data that is relevant to influenza introductions to and spread among poultry populations.

Validity of the findings

no

Additional comments

After the high path outbreak was detected in farms a number of hypotheses were put forth 1) wild birds were spreading and introducing viruses to each farm experiencing outbreaks. 2) viruses were being spread by fomites. 3) viruses were being spread by other minor hosts such as small mammals and song birds. 4) viruses were being spread by humans passively (tools, workers, trucks etc). It's important that we have data so that we can reject those ideas that are not supported in order to properly identify transmission pathways and effective control measures. I am happy that this manuscript attempt to address some of the spurious or unlikely hypotheses. I recommend that the results about the surveillance in waterfowl be either removed or extended. These results are not treated appropriately, nor is their relevance to the rest of the results clear.

·

Basic reporting

This study investigated the role of potential avian and rodent bridge hosts for Avian Influenza viruses (AIV) soon after the outbreak in poultry farms of HPAI H5N2 in 2015 in Iowa, United States. The study design focused on sampling these potential bridge hosts at both wetlands where the AIV reservoir occur (i.e. waterfowls) and at poultry farms where HPAI outbreak happened or could have happened. qPCR and serology diagnostics were implemented on both rodent and bird sampled with no positivity detected in the 449 animal sampled. A point to be noted is that mechanical transmission was investigated by swabbing the external body of animals sampled. The authors also compared the community composition of the samples from the wetland and farm sites to identify which species may be more involved in bridging wetland and farms from an epidemiological point of view.
In the discussion section, these negative results are discussed in relation to positive results found at the same time in waterfowls in the same state. The authors conclude that the species sampled may not be involved in transmitting AIV from waterfowl to poultry farms and that human induced transmission maybe more involved in farm-to-farm transmission.

The article is well written, hypotheses are well stated and tested. The literature references are up to date with the scientific field.

I appreciated the approach and the study design and have only minor comments about the study design and some points in the discussion that could be improved and/or further discussed.

Experimental design

- L162-170: site selection: the logic behind the site selection is not completely stated. You have 2 “pairs” of sites including a wetland and a poultry farm and other sites (wetlands and farms distant from each other). I guess there must be some constraints or other factors that drove site selection but they must be indicated. For example, why choosing the Malcom site outside any “coloured” cell? And why not in the single orange cell?

- L236-247: It is difficult to understand exactly if the waterfowl sampling is part of this study or not. You give a sample size and an AIV prevalence but little other information is available about the sample composition (in terms of species for example). If this information cannot be presented in this manuscript because it was collected by another team, I think this should be clearly stated and only the relevant information should be presented. If the information was collected by the authors, then more information should be given. Did those samples were pooled like the ones for the songbirds?

- L331-337: I think you need to state that you assume that the community sampled is representative of the bird/rodent community present. It is necessary because you make this assumption without stating it and without discussing it in the discussion section (see below).

Validity of the findings

The findings of the study are relevant for the field, well presented (see below a couple of suggestion for improvement) and will inform further AIV wild bird and rodent surveillance in the United States and beyond.

- Figure 1: Maybe use different symbols for wetland and poultry farm sites in order for readers to automatically detect them without referring to the text.

- Table 2: I think it is important to present at least the confidence interval of the prevalence estimation for each species (for the total of each species). Despite all your efforts, the sample size is often low for each species and therefore the confidence that AIV was not present in the species’ population is sometimes weak.

Additional comments

- L.70: but see also a recent publication: Caron et al. 2016 in Journal of Applied Ecology. This publication could be used elsewhere in the manuscript as well (“songbirds have been found to be capable of carrying AIV” – L105-106).

- L.106-108: in case you have not seen them and not to be cited at any cost (and sorry for self-citations but little has been done unfortunately on the topic) but consider also Caron et al. 2009 in Infection, Genetic and Evolution and Caron et al. 2010 in Ecology & Society looking at a conceptual approach and a more risk-based approach respectively to your study design.

- L407-417: I agree that you have identified “potential bridge species” within your sample that can connect wetland and poultry farm. But you should discuss at least 2 points. First, the scale of your study is larger than what I would call an ecosystem scale where you sample birds from adjacent wetlands and farms (a few kilometres). This is the case for two pairs of sites but you analyse your results globally across all sites. So maybe that species detected at one wetland were identical to species detected at a distant farm but not at a close farm. In fact, you assume homogenous distribution of species across your study site and should discuss it further. So the risk of AIV transmission from a wetland to a farm is maybe not “real” at all your sites. Second, you only take into consideration the community of your sample and not the “real” community present at site. This should be stated clearly. Capture techniques, time and season do not randomly sample within the bird community. There is a possibility that your potential bridge species (present and abundant at wetland and farm sites) are not the most abundant and shared species between wetland and farm sites within the “real” wild bird community.

- Given your approach, you still largely sample blindly within the bird/rodent communities. Estimating rodent community is difficult but bird communities can be characterised through bird counts. You should maybe mention that this could be a possibility to improve your protocol if time allows (which can be tricky in the case of an outbreak).

- In the discussion, you should also mention that your species sample size is low for most of the species sampled and therefore that the absence of AIV detection does not prove absence of circulation in those species. L449-452, you mention high sample sizes to reach good estimation of prevalence, but you should also say that at the species level, your sample size is often extremely low and cannot estimate much.
You rightly say that further sampling should target (capture techniques should be adapted) identified potential bridge species with the objective to achieve a better sample size. L413-417: you should specify that more targeted sampling could improve our understanding of some of the potential bridge species identified.

- Finally, in the last section of the discussion, you qualitatively assess the risk of spread of AIV from wildlife to poultry and vice-versa. Maybe you should also emphasize that an outbreak can be triggered by a rare event and then subsequently spread between poultry farms by farm-to-farm contacts. So AIV prevalence in bridge hosts, even if very low, represents a risk with potential high consequences.

- Conclusion: L478-480: can you really say this? I would say that your study did not detect AIV in potential bridge hosts and that it supports the hypothesis that they don’t play a role in AIV epidemiology in Iowa.

---

## Round 0.2 · accepted · Accept

Dear James,

Thank you for your submission to PeerJ. Your manuscript has been accepted for publication in PeerJ. Good Job!

Best regards,

Peirong

Reviewer 1 ·

Basic reporting

No comment

Experimental design

No comment

Validity of the findings

No comment

Additional comments

Great revision; I have no substantive comments or recommended changes; the following comments are meant for discussion only.

Line 214: I don’t think a change is needed, but want to point out that the Spackman et al. paper only compared BHI and PBS, neither with antibiotics which can be important, especially for cloacal/anal swabs which are likely to have strong bacterial communities. I know optimal is in the title of the paper, but this seems like an example of making inference beyond the data set, especially since your samples were not stored in an ultracold prior to shipping to the diagnostic lab.

Line 249: The “As with samples for small birds and mammals,” is a bit misleading because I think the APHIS surveillance uses 3mLs of BHI whereas you used 2mLs. You might just strike the beginning of the sentence or specify the 3mLs (assuming I’m correct about the 3mLs).

Line 277: if you were using an H5N9 isolate, you might mention that the extractions were conducted under enhanced BSL-2 or BSL-3 conditions/in a biosafety cabinet.

Line 285: AB?

Standardize capitalization of influenza: it is only capitalized in Lines 394, Table 3, and Line 411. Also there is a bit of inconsistency in references (some titles are capitalized, but most are not). Also, in Table 3, remove capitalization of Americana for redhead.

Line 415: Although I this is a unreasonable assumption, there are very scant data available on long-term AIV antibody persistence, especially for non-traditional hosts and a single infection. A potential difference between the studies you cite and your study is that geese are long-lived and are likely to have multiple exposures (resulting in an anamnestic response), even within a given year. Multiple exposures increase peak detectable antibody responses and the duration of the response so the 3-6 month estimates may be on the high side for the animals you are sampling.

Line 457 – A very minor point, but the 16% might be biased a bit high. I don’t know if you had access to the sample dates for the APHIS data you used, but the APHIS online data (across the US, not just IA, https://www.aphis.usda.gov/animal_health/downloads/animal_diseases/ai/monthlysummary.pdf) show much higher prevalence in August and September than the months during which you sampled so if you didn’t adjust the prevalence for the time period that overlapped your sampling, the 16% might be on the high side.

Line 496: seems like there might be a missing “to” in this sentence?

·

Basic reporting

Introduction

- L61-63 : first sentence. The impression left by this sentence is that waterfowl are the culprit. Maybe adding something like “(…) can cross the wildlife-domestic animal interface and find optimal conditions to evolve towards higher pathogenicity in poultry production systems , sometimes (…)”.

- L86-87: “most poultry farms now enforce strict biosecurity protocols”: yes… in developed countries. Maybe contextualise here.

- L101-102: I don’t think this sentence is necessary there as you repeat it at the end of the introduction.

Experimental design

- L326-328: are these tests been developed for waterfowl? Are they species sensitive? You should maybe specify this.

Validity of the findings

- The fact that your confidence intervals don’t include zero is a problem for me. You should explain more the reason why, maybe in the M&M section. With only zero results, you cannot have confidence interval above zero, it is impossible within my logic. This could be an artefact of the Bayesian method but at the end I don’t see how all of your results have a non-including zero CI.

- The discussion has been improved taking into account the various pointed out in the previous revision.

Additional comments

Most of my previous comments/remarks have been addressed. I have only a few minor points to be a considered below. I believe the manuscript has reached the standard of publication.